# Effects of a Diabetes Sports Summer Camp on the Levels of Physical Activity and Dimensions of Health-Related Quality of Life in Young Patients with Diabetes Mellitus Type 1: A Randomized Controlled Trial

**DOI:** 10.3390/children10030456

**Published:** 2023-02-26

**Authors:** Lida Skoufa, Eleni Makri, Vassilis Barkoukis, Maria Papagianni, Panagiota Triantafyllou, Evangelia Kouidi

**Affiliations:** 1Laboratory of Human Research and Sport Psychology, Department of Physical Education and Sport Science, Aristotle University of Thessaloniki, 57001 Thessaloniki, Greece; 2Laboratory of Sports Medicine, Department of Physical Education and Sport Science, Aristotle University of Thessaloniki, 57001 Thessaloniki, Greece; 3Department of Nutrition and Dietetics, School of Physical Education, Sport Science and Dietetics, University of Thessaly, 42132 Trikala, Greece; 4Unit of Endocrinology, Diabetes and Metabolism 3rd Department of Pediatrics, School of Health Sciences, Aristotle University of Thessaloniki, Hippokration Hospital of Thessaloniki, 54642 Thessaloniki, Greece; 51st Department of Pediatrics, Aristotle University of Thessaloniki, 54642 Thessaloniki, Greece

**Keywords:** diabetes mellitus type 1, diabetes sports camp, physical activity, health-related quality of life

## Abstract

Physical activity (PA) is considered an important part of the treatment of children with diabetes mellitus type 1 (T1DM). Furthermore, health-related quality of life (HRQoL) affects both the physical and mental health of patients with T1DM. The purpose of the study was to evaluate through a randomized controlled trial the impact of participation in a summer diabetes sports camp on the PA and HRQoL of children and adolescents with T1DM. Eighty-four children and adolescents with T1DM were randomly assigned into an intervention (M = 12.64, SD = 1.82, 30 female) and a control group (M = 12.67, SD = 2.50, 30 female). Intervention group participants attended a ten-day summer diabetes sports camp which included an intensive program of PA (6 h of daily PA), educational and entertaining activities as well as education on the importance of PA in the management of the disease. At baseline and at the end of the study, participants completed measures of physical activity, self-esteem, depression, health status, intention to change behavior, and life satisfaction. Results of the two-way repeated measures analysis showed no statistically significant group differences in PA levels (*p* < 0.05) and HRQoL parameters (*p* < 0.05 for all parameters). In conclusion, the results did not support the effectiveness of a 10-day diabetes sports camp on PA levels and HRQoL for children with T1DM. Longer interventions may be more effective in exerting positive influence on trait parameters of children with T1DM’s quality of life. Participation in such programs on multiple occasions should be evaluated in the future.

## 1. Introduction

T1DM is a chronic metabolic disease that may occur at any age but usually has a juvenile onset and accounts for the 5–10% of diabetic patients [1]. T1DM is characterized as a disease of the developed countries, but there has been an increasing trend in its occurrence over the last three decades worldwide. It has been estimated that worldwide, the prevalence of T1DM is 9.5% [1]. In T1DM diabetes, high blood glucose levels pose serious health risks, and its proper regulation delays or even prevents the appearance of its complications [2].

Research evidence has shown high rates of mental disorders in children and adolescents with diabetes, including anxiety disorders, eating disorders and particularly major depressive disorder, that range from 33–47% [3,4]. The prevalence of depression in children with diabetes is two-fold greater than their peers in the general population, while the prevalence in adolescents is three-fold greater [5]. Evidence has shown that 32% of youth with diabetes experience symptoms of anxiety [6], which has been linked to worse treatment adherence, symptom control and management of the disease [7].

HRQoL is considered to significantly affect physical and mental health as well as the mentality and behavior of patients with T1DM in their everyday life [8]. Almost a decade ago, studies showed that HRQoL of young patients with T1DM was poor compared to their healthy peers, while good metabolic control has been associated with a better QoL [9,10].

According to research evidence, engaging in frequent PA has several health benefits for both healthy and diseased people. Specifically, the evidence that is currently available suggests that PA has a beneficial impact on body composition, functional capacity, insulin sensitivity, glycemic management, glycosylated hemoglobin levels, and stress levels in individuals with T1DM [11,12]. It has been found that only a small percentage of children and adolescents with T1DM take part in exercise programs as part of their leisure and school activities [13]. In addition, previous research has demonstrated that they are unable to reach recommended levels of PA and have lower PA levels when compared to their healthy peers [14]. In order to design exercise programs for children and adolescents with T1DM, diabetes camps have been organized worldwide [15]. These camps aim to promote healthy habits and encourage children with T1DM to adopt an active and healthy lifestyle, and improve their overall HRQoL [15,16].

Researchers have only lately begun to demonstrate the benefits of camps for children and adolescents who suffer from chronic diseases [17], despite the fact that these camps have been around for a very long time. There are contradictory findings in the research about the impact of camps on the HRQoL of children and adolescents with T1DM. Participation in diabetes camps has been shown to positively contribute to the education of young T1DM patients about the disease and its management [17,18]. This, in turn, leads to significant improvements in the glycemic control, self-esteem, and mobility of patients who attend such camps. A recent study [19] showed that although there was a decrease in PA levels one month after participation in an adult diabetes camp, there was a significant increase in a three and six month follow up. This was the case despite the fact that there was a decrease in PA levels one month after participation. A comparative study in children with T1DM who either participated in diabetes camps or attended regular schools showed that the first group met the recommended number of steps per day. In addition, it was discovered that attending diabetes camps offered considerable benefits for health-related issues for children diagnosed with T1DM [20]. On the other hand, participation in diabetes camps did not seem to improve quality of life, anxiety, or psychological status in any of the previous investigations [21,22]. Nevertheless, despite the ambiguous results, there is a general consensus that camp programs for children and adolescents with T1DM offer them the opportunity to interact with their peers in a safe environment while also being supported and supervised by a medical team that specializes in their condition [23]. Furthermore, studies have shown that participation in camps contributes positively to the education of T1DM children and adolescents about the disease and its management, leading to significant improvements in their glycaemic control, their self-esteem and their mobility [18,22].

In general, physical activity has been suggested as an important component of the treatment of children and adolescents with T1DM. Previous research on diabetes camps for young patients has demonstrated that participation is useful in boosting PA, providing benefits for certain quality of life and psychological measures, and increasing the participants’ illness awareness. To the best of our knowledge, no previous research has investigated whether diabetes camps have any positive benefits for young people living with T1DM. In order to fill this gap, the purpose of the present study was to evaluate whether participation of children with T1DM in a short-term summer diabetes sports camp, alongside their healthy peers and under the supervision of a specialized medical team, would assist these children in increasing their levels of PA, thereby improving their HRQoL and psychological status.

## 2. Materials and Methods

### 2.1. Participants

Children attending the Endocrine Unit of the 3rd Pediatric Department of the Aristotle University of Thessaloniki in the Hippokration Hospital of Thessaloniki, outpatient clinics or private physicians’ practices were approached to participate in the study (*n* = 134). In order to determine the adequate sample size that was required to detect our effects, we calculated an a priori power analysis using GPower [24,25]. The F test was selected as the test family, and the repeated measures within subjects ANOVA as the statistical test. Alpha was set to 0.05, and the effect size was set to f = 0.25. The analysis showed that a sample size of 36 participants was required. Participants in the study were required to have been diagnosed with T1DM for at least a year prior to the beginning of the research. Eligible participants ranged in age from 7 to 18 years old. In addition, they should have been maintaining a stable insulin regimen. Exclusion criteria included suffering from another kind of diabetes or another condition that affects ability to exercise (e.g., cardiorespiratory and musculoskeletal diseases). The sample of the study consisted of eighty-four children and adolescents with T1DM who met the eligibility criteria for the study and provided their consent to take part in it. A medical history was obtained from all volunteers, and they also underwent a clinical evaluation and HbA1c measurement in the Diabetes Outpatient Clinic of the 3rd Pediatric Department of the Aristotle University of Thessaloniki. They were all on intensive insulin treatment (a multiple daily insulin injection regimen or continuous subcutaneous insulin infusion via pump). A total of 63 participants were using either continuous or flash glucose monitoring, while the remaining 21 were using a glucose meter. Some 43 participants were on therapy with pens, and 41 were on therapy with pumps. None of the participants had any diabetes-related complications.

### 2.2. Design and Procedure

The study was a randomized controlled trial (Figure 1). After the baseline evaluation, the online statistical computing web program “www.randomizer.org (accessed on 30 June 2021)” was used for the randomization process. Children and adolescents with T1DM were randomly assigned into two groups: the intervention group (*n* = 42), who participated in a 10-day summer diabetes sports camp, and the control group (*n* = 42), who were asked to continue their daily routine without participating in any physical activity intervention.

The study protocol was approved by the Ethics Committee of the School of Physical Education and Sport Science of the Aristotle University of Thessaloniki. After being thoroughly informed about the procedures, all parents gave their written informed consent for their children’s participation in the study.

### 2.3. Measurements

At baseline and at the end of the 10-day study, all participants were asked to complete a battery of questionnaires measuring PA and four parameters of traits of HRQoL: life satisfaction, self-esteem, depression, health status, coping strategies and intention to change health-related behavior. The questionnaires had been used in Greece previously [26].

#### 2.3.1. Physical Activity

Physical activity was measured with the Godin–Shephard Leisure-Time Physical Activity Questionnaire [27], consisting of two questions. The first question assessed the frequency and intensity of the PA in which the respondents participated during the period of a week. More specifically, the frequency of PA was estimated in three types of exercise, intense, moderate and mild, for more than 15 min during leisure time. The second question assessed the frequency of participation (often, sometimes, rarely or never) in any PA in leisure time, during which the participant sweated or his heart “beat” quickly, for a 7-day period.

#### 2.3.2. Life Satisfaction

Life Satisfaction was assessed with the Life Satisfaction Assessment Scale [28] which is designed to measure global cognitive crises of life satisfaction and was developed as a measure of the critical component of subjective well-being [29]. Participants indicated how much they agree or disagree with each of five items (e.g., “I am satisfied with my life”) using a 7-point scale ranging from 7 (I totally agree) to 1 (I totally disagree). In the present study, Cronbach’s alpha was calculated at 0.30. 

#### 2.3.3. Self-Esteem

Self-esteem was measured with the Rosenberg Self-Esteem Scale [30]. The scale consisted of ten items evaluating both positive (e.g., “I feel that I have a number of good qualities”) and negative (e.g., “At times I think I am no good at all”) feelings towards the self. Questions were answered using a 4-point Likert scale from 1 (I strongly disagree) to 4 (I totally agree). Cronbach’s alpha was calculated at 0.76. 

#### 2.3.4. Depression

Depressive symptoms were assessed via the Centre for Epidemiological Studies Depression scale (CES-D) [31]. The scale consists of 20 items estimating the frequency of depressive symptoms during the past week. An example of such one such item is ‘I felt tearful’. This instrument has received support with regard to its validity and reliability in older populations [32]. In the present study, Cronbach’s alpha was calculated at 0.64.

#### 2.3.5. Health Status and Its Treatment and Intent to Change Health-Related Behaviour (HAPA)

The health status assessment, its treatment and the intention to change health-related behavior questionnaire was used [33]. The questionnaire consists of a combination of items classified into three categories. The first category includes seven items measuring health status (e.g., “In general, how would you say your health is?”), changes that have occurred in the health status (e.g., “Compared to my best health status ever, my health in general now is…”) and perceptions about health status in everyday life (e.g., “How much is your everyday life affected by your health?”). Answers were given on 5-point scale, from 1 (bad) to 5 (excellent) for health status and changes in the health status, and on a 5-point scale from 1 (not at all) to 5 (too much) for health status in everyday life. Cronbach’s alpha was calculated at 0.64. The next category concerned action plans for the next period and the adoption of health-promoting behaviors that interpret the participants’ intention to change their eating habits, PA, smoking cessation and participation in medical tests (e.g., “I intend to live a healthier life”). The ten questions were answered on a 7-point scale from 1 (not at all) to 7 (very much). Cronbach’s alpha was calculated at 0.75. The third category evaluated the planning of physical activities and action plans to address any difficulties or obstacles in their implementation. It consisted of nine items (e.g., “I already have concrete plans to exercise”) assessed on a 4-point scale from 1 (not at all) to 4 (very much.). Cronbach’s alpha was calculated at 0.85. The overall high score from the three categories showed a strong intention to change health-related behaviors.

### 2.4. Intervention Design

Children and adolescents diagnosed with T1DM who made up the intervention group attended a 10 day diabetes summer sports camp with their healthy peers. All children took part in the exact same pursuits and shared the exact same environment and sleeping quarters. An intensive program of daily physical activity that included three hours in the morning and three hours in the afternoon was part of the intervention (Table 1). Activities such as swimming, sailing, diving, climbing, sports games, football, tennis volleyball, basketball, handball, ping pong, canoeing, archery, team games, and athletics were among those included in the program. During this period, students also took part in a variety of other events, which included both informative and enjoyable activities (e.g., singing, dancing, daily trips). During the study period, children and adolescents with T1DM had medical supervision. Every day, sessions were held to educate them on the importance of PA for the achievement of good glycaemic control and a better general health status, and the role of a healthy lifestyle in the disease management. Participants in the control group were not subjected to any physical activity intervention. Instead, they were instructed to carry on with their typical day-to-day routines, while also taking a vacation with their parents over the same time period.

### 2.5. Statistical Analysis

The analyses were conducted with SPSS 25.0. A Shapiro–Wilk test was used to verify the normality of the distributions of the parameters under study. The correlations between variables were examined with the Pearson correlation coefficient test. A two-way repeated measures analysis of variance was conducted to test for differences in the dependent variables between the intervention and control groups. A two-tailed *p* value < 0.05 was considered statistically significant.

## 3. Results

### 3.1. Preliminary Analyses

The characteristics of the participants are presented in Table 2. The mean body mass index (BMI) was 20.15 (SD = 3.36) for the intervention group and 20.15 (SD = 4.39) for the control group, indicating a normal weight status. HbA1c levels were above the recommended target for both groups. There were no statistically significant differences (*p* > 0.050) between the two groups in patients’ characteristics, PA and HRQoL indices at baseline. Correlations between the studied variables for the first and second measurement in the total sample are presented in Table 3.

Descriptive statistics of the studied variables in the total sample and the two groups for the two measurements are presented in Table 4. There were no statistically significant differences within groups (first vs. second measurements) and between the two groups in any parameter studied (Table 4).

### 3.2. Effectiveness of the Intervention

The results of the repeated measures analysis for PA did not show a significant interaction (F1,82 = 0.94, *p* > 0.05, *n*^2^ = 0.01) or main effect for time and for group. With respect to the indicators of quality of life, namely satisfaction, depression and self-esteem, no significant interaction (F1,82 = 0.05, *p* > 0.05, *n*^2^ = 0.001 for satisfaction, F1,82 = 2.78, *p* > 0.05, *n*^2^ = 0.03 for self-esteem and F1,82 = 0.06, *p* > 0.05, *n*^2^ = 0.001 for depression) or main effects for time or group were revealed in the analysis.

Regarding the Health Action Process Approach variables, namely health status, changes in health, relationship of health with everyday life, intention, activities planning and planning to address difficulties, the results demonstrated no significant interaction for all variables (F1,82 = 0.95, *p* > 0.05, *n*^2^ = 0.01 for health status, F1,82 = 0.22, *p* > 0.05, *n*^2^ = 0.03 for changes in health, F1,82 = 0.87, *p* > 0.05, *n*^2^ = 0.00 for relation of health with everyday life, F1,82 = 0.40, *p* > 0.05, *n*^2^ = 0.01 for intention, F1,82 = 0.06, *p* > 0.05, *n*^2^ = 0.001 for activities planning and F1,82 = 0.01, *p* > 0.05, *n*^2^ = 0.00 for planning to address difficulties). A statistically significant main effect for the groups emerged only for changes in health (F1,82 = 4.67, *p* < 0.05, *n*^2^ = 0.05) but not for any of the other variables. The analyses also revealed a statistically significant main effect for time only for activity planning (F1,82 = 6.01, *p* < 0.05, *n*^2^ = 0.07), but not for any of the other variables.

## 4. Discussion

This study aimed to examine the effects of a 10-day summer diabetes sports camp on the level of PA and dimensions of HRQoL, in children and adolescents with T1DM. The results of the study indicated that participation in the summer diabetes camp had no impact on the outcome variables.

With respect to the effect of the diabetes sport camp on the PA levels of children and adolescents with T1DM, the findings demonstrated that even though the children in the intervention group received training and information on the role of PA in the regulation of the glycemic index and the improvement of their health status, their actual PA levels did not increase significantly. This was the case despite the fact that the children in the intervention group had attended the diabetes sports camp. This finding is consistent with findings from an earlier study [20] suggesting that participation in camps does not promote immediate increases in levels of PA. On the other hand, in the research conducted by Sikora et al., the positive effects on PA were observed three to six months after the end of the camp. This evidence may suggest a delay in the effects of the camp on the behavior of young people. Another plausible explanation may lie in the fact that the post-intervention survey was administered on the last day of the camp, so it could not capture behavior outside the camp. In the present study, no follow-up was conducted, but it will be important for future studies to investigate whether PA levels may increase after a greater amount of time has passed. 

In addition, there was no discernible change in levels of life satisfaction. Similarly to PA, it is possible that the length of the intervention and the interval between the first and second measurements were responsible for these insignificant results. Furthermore, no effect was observed on young people’s dimensions of HRQoL. This finding contradicts previous findings which showed that participation in such programs led to a significant improvement in children and adolescents’ self-esteem [34,35]. These findings may be attributed to the QoL dimensions measured. To be more specific, low self-esteem and depression are rather stable traits that take a longer period of time to change. This is especially true in situations in which no explicit actions or techniques targeting low self-esteem and/or depression have been encountered, such as in our research, wherein neither of these issues were investigated. We expected that the increase in PA and the socialization among the participants would influence self-esteem and depression. On the other hand, no specific strategies were utilized in order to change either the participants’ levels of depression or self-esteem. Thus, it would appear that tailored techniques may be more effective in trying to generate changes in such stable traits. The incorporation of techniques such as behavioral activation and cognitive re-structuring into the curriculum of the camp might be beneficial to the conduct of subsequent research. 

With respect to intention to change health-related behavior, the null findings of the present study suggest that it is challenging to achieve improvements in HRQoL by participating in a short-period camp. Again, for intentions, specific actions targeting attitudes, norms and self-efficacy are needed [36]. In this case, persuasion, information, modeling, goal setting, social support, and planning are among the strategies that have been found effective in producing more favorable intentions toward healthy behavior [36].

This study is not free of limitations. To begin with, there were no subsequent follow-ups conducted to attest to the long-term effects of the intervention. Past evidence revealed that people’s attitudes and behaviors would be affected in a delayed manner; hence, future research should include such measurements in order to investigate the long-term effects of diabetes camps. Secondly, the quality of life assessment was carried out based on relatively stable characteristics that take a longer amount of time to change. Future studies examining the effect of short interventions would benefit from measuring more situational dimensions of QoL, such as vitality. Nevertheless, the study provides valuable information that can be used in the development of practices aiming to improve T1DM patients’ HRQoL. To be more specific, in order for diabetes camps to be effective, participants need to either attend the camps for longer periods of time or attend multiple camps throughout the year (for example, camps in the fall, winter, and spring) so that improvements in health-related behaviors may occur and eventually become habits [18]. Barone et al. [15] argued that the changes that occur as a result of participation in a camp are not often documented. Nonetheless, life in such camps presents an opportunity to promote PA as a means of treating T1DM [15], develop skills to cope with their disease more effectively, and develop socially and psychologically [16]. In addition, the integration of specific strategies targeting these goals would further increase the effectiveness of diabetes camps in improving T1DM patients’ HRQoL.

## Figures and Tables

**Figure 1 children-10-00456-f001:**
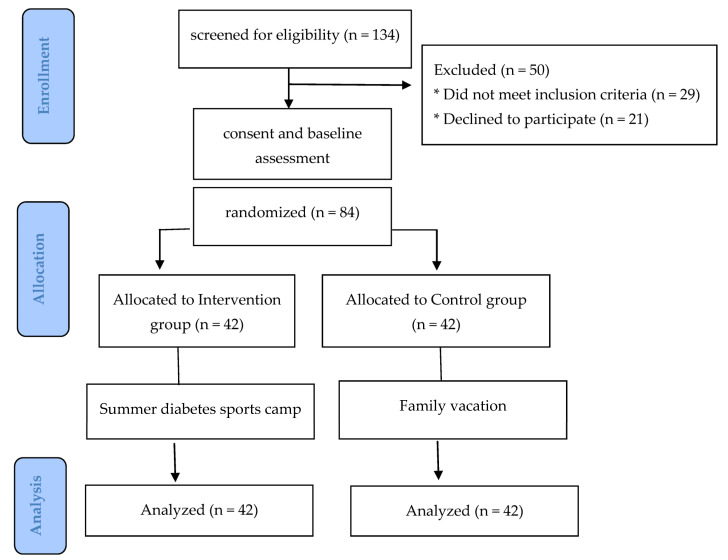
CONSORT diagram of the study design.

**Table 1 children-10-00456-t001:** Description of the intervention.

	Day 1	Day 2	Day 3	Day 4	Day 5	Day 6	Day 7	Day 8	Day 9	Day 10
8:00	Wake-up call	Wake-up call	Wake-up call	Wake-up call	Wake-up call	Wake-up call	Wake-up call	Wake-up call	Wake-up call	Wake-up call
8:30	Breakfast	Breakfast	Breakfast	Breakfast	Breakfast	Breakfast	Breakfast	Breakfast	Breakfast	Breakfast
9:00	Medical supervision	Medical supervision	Medical supervision	Medical supervision	Medical supervision	Medical supervision	Medical supervision	Medical supervision	Medical supervision	Medical supervision
9:30	Education about the importance of PA	Education about the importance of PA	Education about the importance of PA	Education about the importance of PA	Education about the importance of PA	Education about the importance of PA	Education about the importance of PA	Education about the importance of PA	Education about the importance of PA	Education about the importance of PA
10:15	Swimming	Swimming	Swimming	Excursion	Swimming	Swimming	Swimming	Swimming	Swimming	Swimming
11:45	Climbing	Football	Athletics	Excursion	Volleyball	Archery	Ping pong	Basketball	Sport games	Tennis
13:15	Lunch	Lunch	Lunch	Lunch	Lunch	Lunch	Lunch	Lunch	Lunch	Lunch
14:00	Free time	Free time	Free time	Free time	Free time	Free time	Free time	Free time	Free time	Free time
14:45	Rest	Rest	Rest	Rest	Rest	Rest	Rest	Rest	Rest	Rest
16:15	Light meal	Light meal	Light meal	Light meal	Light meal	Light meal	Light meal	Light meal	Light meal	Light meal
16:30	Sailing	Canoe	Diving	Swimming	Seamanship	Sailing	Canoe	Excursion	Swimming	Diving
18:00	Basketball	Tennis	Volleyball	Archery	Handball	Athletics	Football	Excursion	Climbing	Sport games
19:30	Dinner	Dinner	Dinner	Dinner	Dinner	Dinner	Dinner	Dinner	Dinner	Dinner
20:15	Free time	Free time	Free time	Free time	Free time	Free time	Free time	Free time	Free time	Free time
21:00	Singing	Theater	Group activities	Dancing	Singing	Theater	Dancing	Group activities	Singing	Group activities
22:30	Bedtime	Bedtime	Bedtime	Bedtime	Bedtime	Bedtime	Bedtime	Bedtime	Bedtime	Bedtime

**Table 2 children-10-00456-t002:** Demographic and clinical characteristics of the study population (Mean and Standard deviation).

	Total Sample		Intervention Group		Control Group	
	Mean	SD	Mean	SD	Mean	SD
Age (years)	12.65	2.20	12.64	1.82	12.67	2.50
Disease duration (months)	59.05	37.33	63.45	38.03	54.54	36.50
BMI	20.15	3.77	20.15	3.36	20.15	4.39
HbA1c (%)	7.52	1.11	7.53	1.28	7.5	0.94

**Table 3 children-10-00456-t003:** Correlations between variables for the first and second measurement in the total sample. Scores above the diagonal represent the correlations for the first measurement. Scores below the diagonal represent the scores for the second measurement.

	1	2	3	4	5	6	7	8	9	10
1. PA		−0.07	0.04	−0.00	−0.00	0.06	−0.03	0.06	0.12	0.05
2. Satisfaction	0.04		0.21	−0.18	0.31 **	0.03	−0.23 **	−0.01	0.13	0.15
3. Self-esteem	−0.01	0.68 **		−0.43 **	0.36 **	−0.31 **	−0.36 **	−0.12	0.11	0.05
4. Depression	−0.07	−0.46 **	−0.54 **		−0.16	0.15	0.14	0.10	−0.11	−0.00
5. Health status	−0.04	0.59 **	0.48 **	−0.30 **		0.22 *	−0.11	0.22 *	0.03	0.01
6. Changes in health	−0.08	0.04	−0.06	0.01	0.08		0.25 *	0.15	−0.08	0.03
7. Relationship of health with everyday life	0.02	−0.24 *	−0.28 *	0.28 *	−0.18	0.17		−0.15	−0.21	−0.06
8. Intention	−0.08	0.32 **	0.09	0.10	0.30 **	−0.05	−0.16		0.24 *	0.14
9. Activities planning	−0.05	0.06	0.19	−0.01	−0.01	−0.21	−0.20	0.30 **		0.52 **
10. Planning to address difficulties	0.09	−0.01	0.06	0.19	0.07	−0.14	0.04	0.20	0.59 **	

* *p* < 0.05, ** *p* < 0.01.

**Table 4 children-10-00456-t004:** Mean and standard deviation of the studied variables in the total sample and in the two groups.

	Total Sample		Intervention Group		Control Group	
	First Measurement Mean (SD)	Second Measurement Mean (SD)	First Measurement Mean (SD)	Second Measurement Mean (SD)	First Measurement Mean (SD)	Second Measurement Mean (SD)
PA	47.18 (21.99)	49.80 (23.24)	49.45 (25.11)	53.83 (26.65)	44.90 (18.38)	45.76 (18.70)
Satisfaction	5.80 (1.93)	5.65 (1.15)	5.86 (2.55)	5.66 (1.09)	5.74 (1.03)	5.64 (1.22)
Self-esteem	32.33 (4.30)	32.44 (5.16)	31.33 (4.00)	32.12 (4.62)	33.33 (4.39)	32.76 (5.69)
Depression	0.93 (0.32)	0.95 (0.34)	0.91 (0.28)	0.95 (0.32)	0.94 (0.36)	0.96 (0.36)
Health status	3.66 (0.69)	3.62 (0.83)	3.78 (0.64)	3.69 (0.76)	3.54 (0.73)	3.56 (0.90)
Changes in health	2.75 (1.29)	2.74 (1.34)	3.12 (1.27)	2.88 (1.53)	2.38 (1.21)	2.60 (1.34)
Relationship of health with everyday life	2.60 (1.45)	2.62 (1.34)	2.60 (1.53)	2.64 (1.48)	2.60 (1.38)	2.60 (1.21)
Intention	4.78 (1.09)	4.82 (1.27)	4.91 (0.87)	4.89 (1.11)	4.64 (1.27)	4.75 (1.42)
Activities planning	3.16 (0.78)	2.99 (0.90)	3.18 (0.68)	2.09 (0.81)	3.14 (0.87)	2.98 (0.98)
Planning to address difficulties	2.74 (0.66)	2.64 (0.80)	2.79 (0.63)	2.68 (0.78)	2.68 (0.69)	2.59 (0.84)

## Data Availability

The data presented in this study are available on request from the corresponding author.

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
