# Peer review of "Effects of a Diabetes Sports Summer Camp on the Levels of Physical Activity and Dimensions of Health-Related Quality of Life in Young Patients with Diabetes Mellitus Type 1: A Randomized Controlled Trial"

_children, 2023, doi:10.3390/children10030456_

Round 1
Reviewer 1 Report
Tables 1 and 2 show the data for groups A and B, however, in the description of the design and procedures of the study, participants are divided into an intervention group and a control group. It is necessary to use the same terminology.
When describing the groups, it is desirable to provide data on the number of patients on different regimens of therapy (syringe-pens or pumps), CGM or Flashaverage daily doses of insulin, and diabets complications.
Reviewer 2 Report
Title: Effects of a diabetes sports summer camp on the levels of physical activity and dimensions of health-related quality of life in young patients with diabetes mellitus type 1
Article Type: Article
Summary
In this study, the authors examined the effects 10 days sports camp on physical activity and health-related quality of life of children and adolescents with diabetes mellitus type 1. Participants were 84 children and adolescents with T1DM who were randomly assigned into two groups of the 10 days intervention and the control. Dependent variables were measured at baseline as well as at the end of the camp. The results indicated that there is not any significant difference between two groups for all dependent variables.
Evaluation
The topic of this study is interesting for publication in the Journal. However, there are some concerns should be addressed by the authors, in order to improve the quality of the manuscript.
Points and suggestions
Please add an introduction regarding the diabetes and PA and mental health to the abstract and also add the gender of the participants.
Please clarify a little more about the intervention in the abstract.
Please add the significant levels for each comparison in the abstract.
Please add a conclusion to the abstract.
The introduction is good especially in regard to the diabetes and PA relationship. However, I think this is needed the authors clarify more about the diabetes and mental health as well.
How you calculate the sample size?
Please add the ethical code for the study.
What was the initial level of physical fitness and physical activity of the participants?
If your study is a “randomized controlled trial”, you should add this to the title as well as abstract and also you should write the paper according to the CONSORT 2010 guidelines. Because your study is an RCT, it has to be registered in a publicly accessible.
Why did you use the Godin - Shephard Leisure-Time Physical Activity Questionnaire? What was your reason? Why you didn’t use other ones such as the International Physical Activity Questionnaire for Children and Adolescents?
Please add internal consistency (Cronbach's alpha) for all instruments and questionnaires.
I think you should add a table and clarify the intervention in more detail.
In the results section, please add effect size for each comparison.
Round 2
Reviewer 2 Report
Thank you so much. I think this is a good manuscript for publication at the present form.